# Correlations of amide proton transfer-weighted MRI of cerebral infarction with clinico-radiological findings

**Daichi Momosaka[1], Osamu Togao[1]\*, Kazufumi Kikuchi[1], Yoshitomo Kikuchi[1], Yoshinobu Wakisaka[2], Akio Hiwatashi[3]**

**1** Department of Clinical Radiology, Graduate School of Medical Sciences, Kyushu University, Fukuoka, Japan, **2** Department of Medicine and Clinical Science, Graduate School of Medical Sciences, Kyushu University, Fukuoka, Japan, **3** Department of Molecular Imaging & Diagnosis, Graduate School of Medical Sciences, Kyushu University, Fukuoka, Japan

\* togao@radiol.med.kyushu-u.ac.jp

**Data Availability Statement:** All relevant data are within the paper and its Supporting Information files.

## Abstract

### Objective

To clarify the relationship between amide proton transfer-weighted (APTW) signal, which reflects intracellular pH, and clinico-radiological findings in patients with hyperacute to subacute cerebral infarction.

### Materials and methods

Twenty-nine patients (median age, 70 years [IQR, 54 to 74]; 15 men) were retrospectively examined. The 10th, 25th, 50th, 75th, and 90th percentiles of APTW signal ($APT_{10}$, $APT_{25}$, $APT_{50}$, $APT_{75}$ and $APT_{90}$, respectively) were measured within the infarction region-of-interest (ROI), and compared between poor prognosis and good prognosis groups (modified Rankin Scale [mRS] score $\geq 2$ and mRS score <2, respectively). Correlations between APTW signal and time after onset, lesion size, National Institutes of Health Stroke Scale (NIHSS) score, mRS score, and mean apparent diffusion coefficient (ADC) were evaluated.

### Results

The poor prognosis group had lower $APT_{50}$, $APT_{75}$, and $APT_{90}$ than the good prognosis group (–0.66 [–1.19 to –0.27] vs. –0.09 [–0.62 to –0.21]; –0.27 [–0.63 to –0.01] vs. 0.31 [–0.15 to 1.06]; 0.06 [–0.21 to 0.34] vs. 0.93 [0.36 to 1.50] %; p <0.05, respectively). $APT_{50}$ was positively correlated with time after onset (r = 0.37, p = 0.0471) and negatively with lesion size (r = –0.39, p = 0.0388). $APT_{75}$ and $APT_{90}$ were negatively correlated with NIHSS (r = –0.41 and –0.43; p <0.05, respectively). $APT_{50}$, $APT_{75}$ and $APT_{90}$ were negatively correlated with mRS (r = –0.37, –0.52 and –0.57; p <0.05, respectively). $APT_{10}$ and $APT_{25}$ were positively correlated with mean ADC (r = 0.37 and 0.38; p <0.05, respectively).

### Conclusion

We demonstrated correlations between APTW signals of infarctions and clinico-radiological findings in patients with hyperacute to subacute infarctions. The poor prognosis group had a

**Funding:** This work was supported by JSPS KAKENHI grants no. JP17K10410 and no. JP20K08111 (https://www.jsps.go.jp). The funders had no role in study design, data collection and analysis, decision to publish, or preparation of the manuscript.

**Competing interests:** The authors have declared that no competing interests exist.

**Abbreviations:** APTW, amide proton transfer-weighted; ADC, apparent diffusion coefficient; CEST, chemical exchange saturation transfer; CNAWM, contralateral normal-appearing white matter; DWI, diffusion-weighted imaging; IQR, interquartile range; mRS, modified Rankin Scale; $MTR_{asym}$, magnetization transfer ratio asymmetry; NIHSS, National Institutes of Health Stroke Scale; NOE, nuclear Overhauser enhancement; ROI, region-of-interest.

lower APTW signal than the good prognosis group. APTW signal was reduced in large infarctions, infarctions with low ADC, and in patients with high NIHSS and mRS scores.

## Introduction

Ischemic stroke is a leading cause of various neurological sequelae or death throughout the world [1]. A critical reduction of cerebral blood flow leads to a failure in normal cerebral oxygen and glucose metabolism, which causes a shift to anaerobic glycolysis and results in the accumulation of lactic acid and a concomitant decrease in intracellular pH [2]. The intracellular pH of brain tissue varies according to metabolic energy status after stroke [3]. Thus, tissue pH may serve as a potential surrogate biomarker to reflect the metabolic state and tissue damage of ischemic brain tissue [4]. Previous phosphorus-31 MR resonance spectroscopy studies have revealed that early acidosis and subacute alkalosis occurs during ischemic stroke [5, 6]. However, the clinical application of phosphorus-31 MR spectroscopy to brain ischemia is limited by its low spatial and temporal resolution, and the requirement for special coils which are not available in most hospitals [7]. Thus, novel imaging methods to evaluate the changes of tissue pH after ischemic stroke in the clinical setting are required.

Amide proton transfer-weighted (APTW) MRI is a chemical exchange saturation transfer (CEST) imaging method [8]. In the past two decades, APTW MRI has been employed to evaluate the changes of mobile abnormal proteins and peptides in brain tumors [9, 10]. The CEST effect due to amide protons, referred to as the amide proton transfer ratio (APTR), highly depends on tissue pH as shown in the following formula:

$$APTR = \frac{\kappa[amide\ proton]}{2[water\ proton]}(1 - e^{-t_{sat}/T_1})T_1 \tag{1}$$

$$\kappa = 5.57 \times 10^{pH-6.4} \tag{2}$$

where $\kappa$ is proton exchange rate, which is the pH-dependent term as shown in Eq (2); $T_1$ is the T1 relaxation time of water; and $t_{sat}$ is the duration of saturation [11]. Zhou et al. were the first to observe the pH-sensitive amide proton transfer effects in an animal model of brain ischemia in 2003 [12]. Since then, there have been increasing number of animal studies in which pH-sensitive APTW imaging has shown promise in detecting ischemic tissue acidosis following impaired aerobic metabolism [13, 14]. Despite these promising results, its translation to clinical use is still limited, and there have been only a few clinical studies using APTW imaging for ischemic stroke [15–17].

Clarifying the relationship between APTW signal changes and clinical information should enhance our understanding of the pathophysiology of ischemic stroke, and may benefit its future treatment strategies. Therefore, the purpose of this study was to clarify the relationship between APTW imaging characteristics and the clinico-radiological findings in patients with hyperacute to subacute cerebral infarction.

## Materials and methods

### Participants

The institutional clinical research ethics committee of Kyushu University approved the study protocol (IRB Clinical Research number 2019–233). The requirement for informed consent was waived due to the retrospective nature of the study. The inclusion criteria were as follows:

(a) patients who underwent MRI at our institution with clinical suspicions for acute or sub-acute infarctions (<3 weeks) from August 2014 to September 2019. The exclusion criteria were as follows: (a) patients who could not undergo APTW imaging because of clinical demand and availability of MRI unit, (b) lesions less than 10 mm in diameter on the transverse diffusion-weighted image (DWI), (c) insufficient image quality, (d) patient's age <18 years, (e) patients who received endovascular therapy for brain infarction before MRI, (f) hemorrhagic infarction, and (g) other concurrent brain disorders.

Twenty-nine patients (median age, 70 years; interquartile range [IQR], 54 to 74 years) including 15 males (median age, 71 years; IQR, 59 to 75) and 14 females (median age, 63.5 years; IQR, 42.8 to 73.3) met the inclusion and exclusion criteria (Fig 1). In all patients, the clinical subtypes of infarctions were classified according to the TOAST stroke subtype classification system [18]. The time after onset was defined as the duration of symptom onset to MRI. The patients' pre-stroke functional ability was assessed using the pre-stroke modified Rankin Scale (mRS) score [19]. Clinical severity was assessed with the National Institute of Health Stroke Scale (NIHSS) score when patients were admitted to the hospital [20]. The patients' functional outcome was assessed using the mRS score [21] at the end of the follow-up period, at least four weeks after symptom onset. Pre-stroke mRS scores, NIHSS scores on admission, and mRS scores at the chronic phase were determined according to the medical records by a board-certified stroke physician (Y.W., 23 years of experience in stroke medicine).

## MRI data acquisition

MRI was conducted on a 3.0-T MRI system (Achieva TX, Philips Medical Systems, Best, the Netherlands) with an 8-channel head coil.

First, several routine MR sequences including transverse DWI, T2- and T1-weighted images, fluid attenuated inversion recovery, and 3D time-of-flight MR angiography were collected. DWI was performed using a single-shot echo-planar imaging diffusion sequence, with 2 b-values (0, 1000 sec/mm$^2$) in three orthogonal directions. The other imaging parameters were as follows: repetition time = 2,500 ms, echo time = 70 ms, acquisition matrix = $128 \times 126$ (zero filled and reconstructed to $256 \times 256$), field-of-view = $230 \times 230$ mm$^2$, slice thickness = 5 mm, number of slices = 22, resolution = $0.9 \times 0.9 \times 5.0$ mm$^3$, sensitivity encoding factor = 1.5, scan time = 2 min 7 sec.

Patients who underwent MRI at our institution with clinical suspicions for acute or subacute infarctions (<3 weeks) from August 2014 to September 2019 (n = 731)

Excluded (n = 702)

Not underwent amide proton transfer-weighted (APTW) imaging (n = 664)

Lesion size <10 mm (n = 22)

Insufficient image quality (n = 6)

Younger than 18 years old (n = 3)

Endovascular therapy for cerebral thrombosis before MRI (n = 3)

Hemorrhagic infarction (n = 3)

Concurrent other brain disorder (n = 1)

Final enrollment (n = 29)

**Fig 1. Study population flowchart.**

Subsequently, APTW MRI was conducted on the transverse slice with the largest ischemic lesions, which was selected with reference to DWI. APTW MRI was conducted using a single-shot, turbo-spin-echo sequence, with the following parameters: saturation power = 1.5 μT, saturation time = 2.0 sec, repetition time = 5000 ms, echo time = 6 ms, acquisition matrix = 128 × 128, field-of-view = 230 × 230 mm$^2$, slice thickness = 5 mm, resolution = 1.8 × 1.8 × 5.0 mm$^3$. A multi-offset, multi-acquisition APTW MRI protocol was used, and 25 offsets spanned –6 to +6 ppm (25 offsets = 0, ± 0.5, ± 1.0, ± 1.5, ± 2.0, ± 2.5, ± 3.0, ± 3.5, ± 4.0, ± 4.5, ± 5.0, ± 5.5, ± 6.0 ppm). An unsaturated image was also acquired with the pre-saturation pulse set to –1560 ppm for signal normalization. The total duration of the APTW MRI protocol was 2 min 20 sec.

To correct the B0 inhomogeneity of the APTW MRI, a B0 map was acquired separately using the same imaging geometry and spatial resolution as the APTW MRI via a 2D gradient-echo sequence with the following parameters: repetition time = 15 ms, echo time = 8.1 ms, flip angle = 30˚, dual echo (ΔTE = 1 ms), number of acquisitions = 16, scan time = 33 sec.

## Data processing

All image processing was executed using Image J (version 1.48, National Institutes of Health, Bethesda, MD). To assess the APTR, we obtained the magnetization transfer ratio asymmetry ($MTR_{asym}$) on a pixel-by-pixel basis from a Z-spectrum plot of the water signal attenuation versus offset saturation frequency (offset = 0 for water resonance). Because the saturation frequency of amide protons is primarily around +3.5 ppm [12], the APTR was assessed by calculating the $MTR_{asym}$ at ±3.5 ppm according to the following formula [12]:

$$MTR_{asym}(3.5\ ppm) = S_{sat}(-3.5\ ppm)/S_0 - S_{sat}(+3.5\ ppm)/S_0$$
$$= APTR + MTR'_{asym}(3.5\ ppm) \tag{3}$$

where $S_{sat}$ and $S_0$ were the signal intensities obtained with and without selective radiofrequency saturation pulse irradiation, respectively. $MTR'_{asym}(3.5\ ppm)$ represents the upfield nuclear Overhauser enhancement (NOE) effect of various non-exchangeable protons [22].

Eq (3) shows that $MTR_{asym}(3.5\ ppm)$ contains not only the APTR, but also contributions from the NOE effect. Thus, we refer to $MTR_{asym}(3.5\ ppm)$ as the APTW signal (%), and $MTR_{asym}(3.5\ ppm)$ maps as APTW images.

## Qualitative analysis of APTW images

As a qualitative analysis, the visibility of infarctions on APTW images was independently evaluated by two board-certified neuroradiologists (observer 1, D.M., 5 years of experience; observer 2, O.T., 19 years of experience) using a three-level visual grading scale as follows: clear, the infarction can be clearly identified as a hypointense lesion on an APTW image without reference to DWI; moderate, an infarction can be recognized as a hypointense lesion on APTW images, but it is difficult to identify the lesion without reference to DWI; unclear, a lesion cannot be recognized on an APTW image even with reference to DWI.

The inter-rater agreement for the visibility between the two observers was evaluated using Cohen's kappa coefficient [23].

## Quantitative analysis of APTW images

The quantitative analysis was performed by a board-certified neuroradiologist (D.M., 5 years of experience). A region-of-interest (ROI) was manually drawn to include an entire hyperintense lesion on DWI; this ROI was then copied and pasted onto the APTW image. Another ROI was placed on the contralateral normal-appearing white matter (CNAWM).

We obtained the 10th, 25th, 50th, 75th, and 90th percentiles of the histogram for the APTW signal (%) within the ROIs ($APT_{10}$, $APT_{25}$, $APT_{50}$, $APT_{75}$, and $APT_{90}$, respectively); the lowest 10% of the signal within an ROI was found below $APT_{10}$, and the highest 10% was found above $APT_{90}$. Histogram analysis is a quantitative technique has been used in previous CEST studies, e.g. Liu et al. applied a histogram distribution-based analysis for CEST signals of a mouse model of multiple sclerosis [24].

The mean apparent diffusion coefficient (ADC) ($mm^2/sec$) within the infarction ROI was also obtained. Lesion size was measured as the maximum diameter of the infarction on the transverse DWI.

For the qualitative analyses, the APTW signal percentiles within the infarction ROI were compared to those within the CNAWM ROI using the Wilcoxon signed-rank test. Demographic and baseline clinical characteristics and the APTW signal percentiles within the infarction ROI were compared between cardioembolic infarctions and non-cardioembolic infarctions, and between the poor prognosis group (mRS score $\geq 2$) and the good prognosis group (mRS score $<2$), using the Chi-squared and Mann–Whitney U tests. The correlations between the APTW signal percentiles within the infarction ROI and the time after onset, lesion size, NIHSS score on admission, mRS score at the chronic phase, and mean ADC within the infarction ROI were evaluated using Spearman's rank correlation coefficient.

The statistical analyses were performed using JMP Pro 14.2.0 (SAS Institute, Cary, NC) under supervision of a statistician (J.K.). P-values $<0.05$ was considered statistically significant.

## Results

### Demographic and baseline clinical characteristics

The infarction etiologies were as follows: cardioembolic infarction, n = 10; non-cardioembolic infarction, n = 19 (large-artery atherosclerosis, n = 6; small vessel occlusion, n = 1; other determined etiology, n = 9; undetermined etiology, n = 3). Twenty-seven patients had infarctions in the cerebral hemispheres and two patients had infarctions in the cerebellar hemispheres. The median time after onset was 52.0 hours [15.3 to 129.0] (median [IQR]) (hyperacute infarction, 1.5–17 hours after onset, n = 9; acute infarction, 31–136 hours after onset, n = 15; subacute infarction, 189–322 hours after onset, n = 5). The pre-stroke mRS score was 1.0 [0 to 2.5], and the median NIHSS score on admission was 4.0 [2.0 to 10.5]. The mRS score at the chronic phase of stroke was 3.0 [1.0 to 4.5] (good prognosis, 0–1, n = 8; poor prognosis, 2–6, n = 21) with the follow-up period of 90.0 [70.5 to 90.0] days.

### Qualitative analysis of APTW images

Observer 1 judged seven (24.1%), ten (34.5%), and twelve lesions (41.4%) as clear, moderate, and unclear, respectively. Observer 2 judged seven (24.1%), eleven (37.9%), and eleven lesions (37.9%) as clear, moderate, and unclear, respectively. The inter-rater agreement was moderate (kappa value = 0.84, p $<0.001$).

### Quantitative analysis of APTW images

The APTW signal percentiles within the infarction and CNAWM ROIs are presented in Table 1. The $APT_{10}$, $APT_{25}$, and $APT_{50}$ of the infarctions were significantly lower than those of the CNAWM (median $APT_{10}$, –1.41 [IQR, –1.96 to –0.73] vs. –0.72 [–1.32 to –0.32] %, p $<0.0001$; $APT_{25}$, –0.94 [–1.53 to –0.45] vs. –0.50 [–0.97 to –0.11] %, p = 0.0005; $APT_{50}$, –0.53 [–1.18 to –0.13] vs. –0.24 [–0.64 to 0.16] %, p = 0.0027). The median lesion size was 46.0 mm

**Table 1. Percentiles of APTW signal of infarction and CNAWM (n = 29).**

|  | Infarction | CNAWM | p-value |
|---|---|---|---|
| $APT_{10}$, % | −1.41 (−1.96 to −0.73) | −0.72 (−1.32 to −0.32) | < 0.0001* |
| $APT_{25}$, % | −0.94 (−1.53 to −0.45) | −0.50 (−0.97 to −0.11) | 0.0005* |
| $APT_{50}$, % | −0.53 (−1.18 to −0.13) | −0.24 (−0.64 to 0.16) | 0.0027* |
| $APT_{75}$, % | −0.18 (−0.47 to 0.27) | −0.01 (−0.44 to 0.41) | 0.1739 |
| $APT_{90}$, % | 0.15 (−0.11 to 0.60) | 0.29 (−0.23 to 0.65) | 0.8085 |

Data are expressed as median (interquartile range). $APT_{10}$, $APT_{25}$, $APT_{50}$, $APT_{75}$, and $APT_{90}$ are the 10th, 25th, 50th, 75th, and 90th percentiles of the APTW signal within the infarction and CNAWM ROIs, respectively.
* indicates statistically significant (p <0.05). APTW, amide proton transfer-weighted; CNAWM, contralateral normal-appearing white matter; ROI, region-of-interest

[IQR, 32.5 to 71.5] (17–24 mm, n = 6; 25–49 mm, n = 10; 50–74 mm, n = 8; 75–99 mm, n = 2; 100–132 mm, n = 3).

The baseline clinical characteristics and the percentiles in the cardioembolic and other sub-types of cerebral infarction are presented in Table 2. There were no significant differences in the clinical characteristics and APTW signals within the infarction ROI between the two groups (p >0.05).

Table 3 shows the baseline clinical characteristics and percentiles of the two groups divided according to their mRS scores at the chronic period. The baseline clinical characteristics were not significantly different between the poor and good prognosis groups. The $APT_{50}$, $APT_{75}$, and $APT_{90}$ within the infarction ROI were significantly lower in the poor prognosis group than those in the good prognosis group (median $APT_{50}$, −0.66 [IQR, −1.19 to −0.27] vs. −0.09 [−0.62 to −0.21] %, p = 0.0481; $APT_{75}$, −0.27 [−0.63 to −0.01] vs. 0.31 [−0.15 to 1.06] %, p = 0.0104; $APT_{90}$, 0.06 [−0.21 to 0.34] vs. 0.93 [0.36 to 1.50] %, p = 0.0037).

The correlations between APTW signal within the infarction ROI and time after onset, lesion size, NIHSS score, mRS score, and mean ADC are summarized in Table 4. The $APT_{50}$ was positively correlated with time after onset (r = 0.37, p = 0.0471), and inversely correlated with lesion

**Table 2. Demographic and baseline clinical characteristics and percentiles of APTW signal of cardioembolic (n = 10) and non-cardioembolic (n = 19) infarctions.**

|  | Cardioembolic | Non-cardioembolic | p-value |
|---|---|---|---|
| Gender, male:female | 7: 3 | 8: 11 | 0.1603 |
| Age, yrs | 66 (43.3 to 76.3) | 71 (55.0 to 74.0) | 0.8363 |
| Time after onset, hrs | 33.6 (4.1 to 113.1) | 70.3 (31.7 to 129.3) | 0.3352 |
| Lesion size, mm | 55.5 (20.0 to 83.3) | 42 (35.0 to 57.0) | 0.4626 |
| Pre-stroke mRS score | 0.5 (0 to 2.0) | 1.0 (0 to 3.0) | 0.3142 |
| NIHSS score | 4 (2.5 to 15.5) | 3 (2.0 to 7.0) | 0.5182 |
| mRS score | 1.5 (0 to 3.3) | 4 (2.0 to 6.0) | 0.0562 |
| $APT_{10}$, % | −1.62 (−2.38 to −0.78) | −1.18 (−1.88 to −0.69) | 0.3019 |
| $APT_{25}$, % | −1.11 (−1.63 to −0.51) | −0.89 (−1.53 to −0.41) | 0.5663 |
| $APT_{50}$, % | −0.53 (−1.27 to −0.18) | −0.53 (−1.18 to −0.07) | 0.6300 |
| $APT_{75}$, % | −0.09 (−0.59 to 0.37) | −0.18 (−0.44 to 0.21) | 0.9451 |
| $APT_{90}$, % | 0.43 (−0.29 to 1.16) | 0.14 (−0.05 to 0.54) | 0.6629 |

Data are expressed as median (interquartile range). $APT_{10}$, $APT_{25}$, $APT_{50}$, $APT_{75}$, and $APT_{90}$ are the 10th, 25th, 50th, 75th, and 90th percentiles of APTW signal within the infarction ROI, respectively. APTW, amide proton transfer-weighted; mRS, modified Rankin Scale; NIHSS, National Institutes of Health Stroke Scale; ROI, region-of-interest

**Table 3. Demographic and baseline clinical characteristics and percentiles of APTW signal of the poor prognosis group with mRS score ≥2 (n = 21) and good prognosis group with mRS score <2 (n = 8).**

| | Poor prognosis | Good prognosis | p-value |
|---|---|---|---|
| Gender, male:female | 10: 11 | 5: 3 | 0.4812 |
| Age, yrs | 72.0 (56.0 to 75.5) | 61.0 (47.5 to 70.8) | 0.1240 |
| Time after onset, hrs | 41.2 (15.3 to 102.5) | 100.8 (19.7 to 177.6) | 0.3171 |
| Lesion size, mm | 48.0 (36.0 to 72.0) | 37.0 (22.5 to 67.3) | 0.3536 |
| Pre-stroke mRS score | 1.0 (0 to 3.0) | 0.5 (0 to 1.8) | 0.2731 |
| NIHSS score | 5.0 (2.5 to 11.0) | 3.0 (1.0 to 10.8) | 0.2107 |
| Cardioembolic:Non-cardioembolic | 5: 16 | 5: 3 | 0.0834 |
| $APT_{10}$, % | −1.41 (−2.12 to −0.83) | −1.12 (−1.78 to −0.53) | 0.2941 |
| $APT_{25}$, % | −1.12 (−1.62 to −0.59) | −0.62 (−1.09 to −0.21) | 0.1367 |
| $APT_{50}$, % | −0.66 (−1.19 to −0.27) | −0.09 (−0.62 to 0.21) | 0.0481* |
| $APT_{75}$, % | −0.27 (−0.63 to −0.01) | 0.31 (−0.15 to 1.06) | 0.0104* |
| $APT_{90}$, % | 0.06 (−0.21 to 0.34) | 0.93 (0.36 to 1.50) | 0.0037* |

Data are expressed as median (interquartile range). $APT_{10}$, $APT_{25}$, $APT_{50}$, $APT_{75}$, and $APT_{90}$ are the 10th, 25th, 50th, 75th, and 90th percentiles of APTW signal within the infarction ROI, respectively.

* indicates statistically significant ($p < 0.05$). APTW, amide proton transfer-weighted; mRS, modified Rankin Scale; NIHSS, National Institutes of Health Stroke Scale

size (r = −0.39, p = 0.0388). $APT_{75}$ and $APT_{90}$ were inversely correlated with the NIHSS score ($APT_{75}$, r = −0.41, p = 0.0261; $APT_{90}$, r = −0.43, p = 0.0189). $APT_{50}$, $APT_{75}$, and $APT_{90}$ were inversely correlated with the mRS score at the chronic phase of stroke ($APT_{50}$, r = −0.37, p = 0.0451; $APT_{75}$, r = −0.52, p = 0.0035; $APT_{90}$, r = −0.57, p = 0.0013). $APT_{10}$ and $APT_{25}$ were positively correlated with mean ADC ($APT_{10}$, r = 0.37, p = 0.0457; $APT_{25}$, r = 0.38, p = 0.0439).

## Sensitivity analysis of study participants without pre-stroke dependency

The sensitivity analysis of a subpopulation of patients without pre-stroke dependency (n = 18, pre-stroke mRS≦1) showed the same trend for an association between the APTW signal within the infarction ROI and the stroke outcome (see details in S1 Appendix): the $APT_{50}$, $APT_{75}$, and $APT_{90}$ were significantly lower in the poor prognosis group than those in the good prognosis group (median $APT_{50}$, −0.74 [IQR, −1.42 to −0.35] vs. 0 [−0.44 to 0.42] %, p = 0.0169; median $APT_{75}$, −0.36 [−0.82 to −0.07] vs. 0.45 [−0.06 to 1.44] %, p = 0.0131; median $APT_{90}$, 0.03 [−0.44 to 0.49] vs. 0.94 [0.21 to 2.21] %, p = 0.0169). Moreover, $APT_{50}$, $APT_{75}$, and $APT_{90}$ were inversely correlated with the mRS score at the chronic phase of stroke ($APT_{50}$, r = −0.54, p = 0.0211; $APT_{75}$, r = −0.59, p = 0.0099; $APT_{90}$, r = −0.57, p = 0.0130).

Figs 2–4 show representative cases of cerebral infarctions.

**Table 4. Correlations between percentiles of APTW images and clinico-radiological findings in infarctions (n = 29).**

| | Time after onset, hrs | Lesion size, mm | NIHSS score | mRS score | Mean ADC, mm²/sec |
|---|---|---|---|---|---|
| $APT_{10}$, % | r = 0.16, p = 0.3628 | r = −0.27, p = 0.1522 | r = −0.14, p = 0.4753 | r = −0.16, p = 0.3939 | r = 0.37, p = 0.0457* |
| $APT_{25}$, % | r = 0.24, p = 0.2139 | r = −0.36, p = 0.0552 | r = −0.23, p = 0.2220 | r = −0.25, p = 0.1846 | r = 0.38, p = 0.0439* |
| $APT_{50}$, % | r = 0.37, p = 0.0471* | r = −0.39, p = 0.0388* | r = −0.33, p = 0.0811 | r = −0.37, p = 0.0451* | r = 0.33, p = 0.0838 |
| $APT_{75}$, % | r = 0.34, p = 0.0692 | r = −0.33, p = 0.0809 | r = −0.41, p = 0.0261* | r = −0.52, p = 0.0035* | r = 0.27, p = 0.1528 |
| $APT_{90}$, % | r = 0.28, p = 0.1375 | r = −0.22, p = 0.2427 | r = −0.43, p = 0.0189* | r = −0.57, p = 0.0013* | r = 0.08, p = 0.6826 |

$APT_{10}$, $APT_{25}$, $APT_{50}$, $APT_{75}$, and $APT_{90}$ are the 10th, 25th, 50th, 75th, and 90th percentiles of the APTW signal within the infarction ROI, respectively.

* indicates statistically significant ($p < 0.05$). APTW, amide proton transfer-weighted; ADC, apparent diffusion coefficient; NIHSS, National Institutes of Health Stroke Scale; mRS, modified Rankin Scale; ROI, region-of-interest

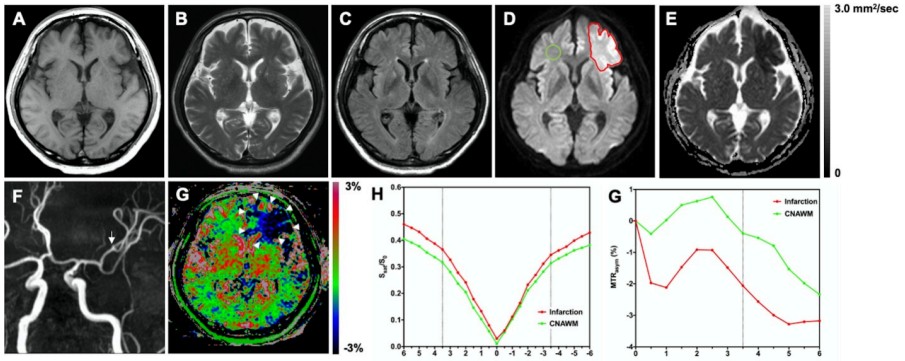

**Fig 2. A 60-year-old male with a hyperacute cardioembolic infarction fifteen hours after the onset.** The pre-stroke mRS score was one. The NIHSS score at the time of admission was 17. The mRS score at 90 days after the onset was one. No signal abnormality is seen on **A:** T1-weighted image, **B:** T2-weighted image, and **C:** FLAIR image. **D:** DWI shows the hyperintense infarct area in the left frontal lobe. The ROI of the infarct area (red line) and the CNAWM (green line) is indicated. **E:** The infarct area shows restricted diffusivity (mean ADC = $0.41 \times 10^{-3}$ mm$^2$/sec). **F:** MR angiography shows the occlusion of the frontal branch of the left middle cerebral artery (arrow). **G:** The infarct area was hypointense on the APTW image (APT$_{10}$ = −3.52%, APT$_{25}$ = −2.90%, APT$_{50}$ = −2.13%, APT$_{75}$ = −0.22%, APT$_{90}$ = 1.50%) (arrowheads). On the visual grading scale, this case was graded as clear by both raters. **H:** Z-spectra show a lower CEST effect in the entire offset range (−6 to +6 ppm) in the infarct area compared to the CNAWM. **G:** MTR$_{asym}$ spectra show decreased MTR$_{asym}$ (3.5ppm) or APTW signal (%) in the infarct area compared to the CNAWM. mRS, modified Rankin Scale; NIHSS, National Institutes of Health Stroke Scale; FLAIR, fluid attenuated inversion recovery; DWI, diffusion-weighted imaging; ROI, region-of-interest; ADC, apparent diffusion coefficient; APTW, amide proton transfer-weighted; CEST, chemical exchange saturation transfer; CNAWM, contralateral normal-appearing white matter; S$_{sat}$(ppm) and S0, the signal intensities obtained with and without selective radiofrequency saturation pulse irradiation, respectively; APT$_{10}$, APT$_{25}$, APT$_{50}$, APT$_{75}$, and APT$_{90}$ correspond to the 10th, 25th, 50th, 75th, and 90th percentiles of the APTW signal value within the ROI, respectively.

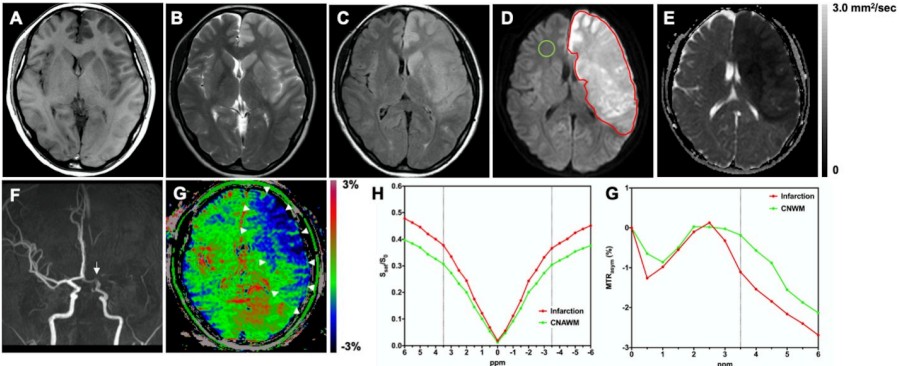

**Fig 3. A 27-year-old female with an acute cardioembolic infarction fifteen hours after the onset.** The pre-stroke mRS score was zero. The NIHSS score at the time of admission was 15. The mRS score at 90 days after the onset was four. The infarct area in the left cerebral hemisphere is hypointense on **A:** T1-weighted image; and hyperintense on **B:** T2-weighted image, **C:** FLAIR, and **D:** DWI. The ROI of the infarct area (red line) and the CNAWM (green line) is indicated. **E:** The infarct area shows restricted diffusivity (mean ADC = $0.36 \times 10^{-3}$ mm$^2$/sec). **F:** MR angiography shows the occlusion of the left middle cerebral artery (arrow). **G:** The infarct area was hypointense on the APTW image (APT$_{10}$ = −2.29%, APT$_{25}$ = −1.93%, APT$_{50}$ = −1.51%, APT$_{75}$ = −1.10%, APT$_{90}$ = −0.69%) (arrowheads). This case was graded as clear by both raters according to the visual grading scale. **H:** Z-spectra show a lower CEST effect in entire offset range (−6 to +6 ppm) in the infarct area compared to the CNAWM. **G:** MTR$_{asym}$ spectra show decreased MTR$_{asym}$(3.5ppm) or APTW signal (%) in the infarct area compared to the CNAWM. mRS, modified Rankin Scale; NIHSS, National Institutes of Health Stroke Scale; FLAIR, fluid attenuated inversion recovery; DWI, diffusion-weighted imaging; ROI, region-of-interest; ADC, apparent diffusion coefficient; APTW, amide proton transfer-weighted; CEST, chemical exchange saturation transfer; CNAWM, contralateral normal-appearing white matter; S$_{sat}$(ppm) and S$_0$, the signal intensities obtained with and without selective radiofrequency saturation pulse irradiation, respectively; APT$_{10}$, APT$_{25}$, APT$_{50}$, APT$_{75}$, and APT$_{90}$, correspond to the 10th, 25th, 50th, 75th, and 90th percentiles of APTW signal value, respectively.

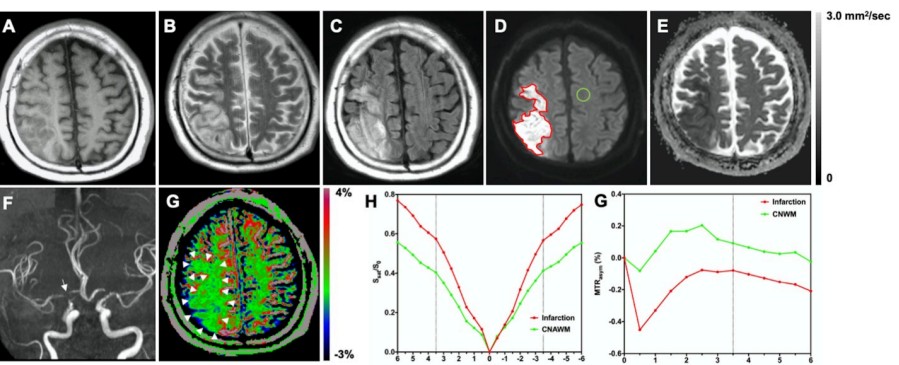

**Fig 4. A 59-year-old male with a subacute atherosclerotic infarction nine days after the onset.** The pre-stroke mRS score was zero. The NIHSS score at the time of admission was 10. The mRS score at 90 days after the onset was four. The infarct area in the right cerebellar hemisphere is hypointense on **A:** T1-weighted image; and hyperintense on **B:** T2-weighted image, **C:** FLAIR, and **D:** DWI. The ROI of the infarct area (red line) and the CNAWM (green line) are indicated. **E:** The infarct area shows restricted diffusivity (mean ADC = $0.51 \times 10^{-3}$ mm²/sec). **F:** MR angiography shows the terminus stenosis of the right internal carotid artery (arrow). **G:** The infarct area was relatively hypointense on the APTW image ($APT_{10}$ = −0.68%, $APT_{25}$ = −0.41%, $APT_{50}$ = −0.07%, $APT_{75}$ = 0.30%, $APT_{90}$ = 0.64%) (arrowheads). This case was graded as moderate by both raters on the visual grading scale. **H:** Z-spectra show a lower CEST effect in the entire offset range (−6 to +6 ppm) in the infarct area compared to the CNAWM. **G:** $MTR_{asym}$ spectra show decreased $MTR_{asym}$(3.5ppm) or APTW signal (%) in the infarct area compared to the CNAWM. mRS, modified Rankin Scale; NIHSS, National Institutes of Health Stroke Scale; FLAIR, fluid attenuated inversion recovery; DWI, diffusion-weighted imaging; ROI, region-of-interest; ADC, apparent diffusion coefficient; APTW, amide proton transfer-weighted; CEST, chemical exchange saturation transfer; CNAWM, contralateral normal-appearing white matter; $S_{sat}$ (ppm) and $S_0$, the signal intensities obtained with and without selective radiofrequency saturation pulse irradiation, respectively; $APT_{10}$, $APT_{25}$, $APT_{50}$, $APT_{75}$, and $APT_{90}$, correspond to the 10th, 25th, 50th, 75th, and 90th percentiles of APTW signal value, respectively.

## Discussion

In this study, we observed a significantly reduced APTW signal in the infarct area compared to that in the CNAWM, in agreement with previous studies [12, 13, 17].

As shown in Eqs (1) and (2), the APTR depends on not only tissue pH, but also T1 relaxation time, water content, and amide proton content. However, it has been previously reported that APTR changes are primarily determined by pH changes [12]. In addition, a previous study using phosphorus-31 MR spectroscopy has demonstrated that pH abnormalities of the infarct area remain, even in the subacute phase [6]. T1 relaxation time and water content of the infarct area have been shown to both increase over time during the acute to subacute phase due to cytotoxic edema [25]; as the T1 relaxation time has a strong positive correlation with the water content in the infarct area [26], the two changes may cancel each other out to some extent. There has been no previous study focusing on dynamic changes of the amide proton content within the infarction. However, based on a study investigating amide proton content in peritumoral brain edema [27], we speculate that amide proton content might decline in infarctions with cytotoxic edema, which would affect the APTR.

As shown in Eq (3), the APTW signal includes not only the APTR but also $MTR'_{asym}$(3.5 ppm), which is attributed to the upfield NOE effect [22]. However, a recent quantitative CEST study using animal models confirmed that the APTR accounts for a greater part of the $MTR_{asym}$(3.5 ppm) changes than the NOE effect [28]. Collectively, this suggests that the APTW signal would primarily depend on tissue pH of the infarction, even in the subacute phase. Therefore, the reduction in APTW signal in the infarction observed in the present study likely reflects the reduced CEST effect of amide protons due to pH reduction.

The APTW signal within the infarction ROI was significantly lower in the poor prognosis group than in the good prognosis group, which was similar to the findings of Lin et al. [29].

Although infarct volume could be a prognostic factor for patients with cerebral infarction [30], there were no significant differences in the lesion size between the two groups. Thus, the APTW signal of the infarct area might be a new prognostic factor, independent from a lesion size.

Our analyses showed that the APTW signal within the infarction was positively correlated with the time after onset. This might reflect the alleviation of tissue acidification that occurs over time after the onset, as previously demonstrated in both animal [13] and clinical [17] studies. This result is consistent with previous studies using phosphorus-31 MR spectroscopy [5, 6]. The increase in pH after stroke could be explained by active compensatory mechanisms within the ischemic tissue, such as microglial invasion and alternate buffering mechanisms [6].

The APTW signal within the infarction was inversely correlated with the lesion size. We assume that larger infarctions could have a smaller surface area to volume ratio, which would reduce gas-diffusion from surrounding area and result in severe metabolic dysfunction.

The APTW signal within the infarction was significantly inversely correlated with both the NIHSS score on admission and the mRS score at the chronic phase. Lin et al. [29] have previously shown that the difference of the APTW signal between the acute ischemic region and the contralateral side is linearly correlated with both the NHISS score and mRS score. Thus, these results indicate that the APTW signal may be a useful tool to assess stroke severity and predict outcome.

A previous study has demonstrated that a high pre-stroke mRS score is significantly associated with poor functional outcomes at the chronic phase [19]. Thus, we were concerned that the pre-stroke mRS score could be a confounding factor when evaluating the relationship between the APTW signal and mRS score at the chronic phase. Therefore, we conducted an additional sensitivity analysis of a patient subpopulation comprising only those with low pre-stroke mRS scores (pre-stroke mRS ≤1), to exclude the influence of a high pre-stroke mRS score. This sensitivity analysis showed that APTW signal within the infarction in the poor prognosis group was significantly lower than that in the good prognosis group, and was inversely correlated with the mRS score at the chronic phase, which agreed with the main results. These findings suggest that a low APTW signal within the infarction could predict poor outcome, regardless of pre-stroke mRS score.

We found that the APTW signal was positively correlated with ADC in infarctions, indicating that the pH reduction may correlate with the severity of cytotoxic edema. One explanation for this is that acid-sensing ion channels, which are hydrogen ion-gated cation channels that are permeable to $Na^+$, are activated as pH falls but are inactive at physiological pH [31]. Thus, pH reduction would promote intracellular $Na^+$ accumulation via the activation of those channels, worsening the cytotoxic edema.

The two observers judged 11 and 12 out of the 29 infarctions as unclear on the visual grading scale. This may reflect an insufficient sensitivity to pH changes in the protocol used in this study. Concomitantly, undetected lesions might have less tissue acidosis compared to the detected lesions. Optimization of the acquisition parameters, application of a fitting method [32], correction with T1-relaxation time [33, 34] and reduction of motion effect [35] would be helpful to improve the sensitivity of APTW images to tissue pH.

There were no significant differences in the APTW signals within the infarctions between cardioembolic infarctions and the other subtypes. The pH reduction might be less influenced by clinical subtypes compared to other factors such as lesion size or time after onset. Since the number of patients in each group was small, the effect of clinical subtype should be evaluated in a larger patient population.

Our study had several limitations. First, only a limited number of the records of patients with suspected acute infarctions included APTW MRI. When patients are treated for

infarctions, the opportunity to perform new treatment and assessment protocols is limited. Therefore, the sample size was small (n = 29) in this study period. Second, although we excluded cases with intense participant motion, it remains possible that participant motion negatively affected the image quality. Third, we determined pre-stroke mRS scores, NIHSS scores on admission, and mRS scores at the chronic phase retrospectively with a stroke physician supervising to avoid inaccurate scoring. Fourth, to date, the optimal saturation pulse strength for APTW imaging has not been established, and saturation pulse strengths of 0.5–3 μT have been used in previous studies [14–16, 36, 37]. We selected a saturation power of 1.5 μT based on a previous animal study that reported the relationship between APTW signal and tissue lactic acidosis [14]. However, a human study has demonstrated that a saturation power of either 2 or 3 μT can result in a better APTW image contrast between the infarction and the CNAWM than that of 1 μT [15]. Considering these facts, we feel that the saturation pulse of 1.5 μT used in the present study was acceptable; however, slightly stronger saturation powers might have improved the APTW image contrast.

In conclusion, we demonstrated correlations between the APTW signal of infarctions and the clinico-radiological findings in patients with hyperacute to subacute infarctions. The poor prognosis group had a lower APTW signal than good prognosis group. APTW signal was reduced in large infarctions, infarctions with low ADC, and in patients with high NIHSS and mRS scores.

## Supporting information

**S1 Appendix. Supporting methods and results.**
(DOCX)

## Acknowledgments

The authors thank Junji Kishimoto, PhD, from the Center for Clinical and Translational Research, Kyushu University Hospital, for his contribution to the statistical analysis.

## Author Contributions

**Conceptualization:** Daichi Momosaka, Osamu Togao.

**Data curation:** Daichi Momosaka.

**Formal analysis:** Daichi Momosaka.

**Investigation:** Daichi Momosaka, Osamu Togao.

**Methodology:** Daichi Momosaka, Osamu Togao.

**Project administration:** Daichi Momosaka.

**Resources:** Daichi Momosaka, Kazufumi Kikuchi, Yoshitomo Kikuchi, Akio Hiwatashi.

**Software:** Daichi Momosaka.

**Supervision:** Osamu Togao, Kazufumi Kikuchi, Yoshinobu Wakisaka, Akio Hiwatashi.

**Validation:** Osamu Togao.

**Visualization:** Daichi Momosaka.

**Writing – original draft:** Daichi Momosaka.

**Writing – review & editing:** Osamu Togao, Kazufumi Kikuchi, Yoshitomo Kikuchi, Yoshinobu Wakisaka, Akio Hiwatashi.

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
