## [Decision Letter · Decision Letter 0]

21 May 2020

PONE-D-20-12833

Correlations of Amide Proton Transfer-Weighted MRI of Cerebral Infarction with Clinico-radiological Findings

PLOS ONE

Dear Dr. Togao,

Thank you for submitting your manuscript to PLOS ONE. After careful consideration, we feel that it has merit but does not fully meet PLOS ONE’s publication criteria as it currently stands. Therefore, we invite you to submit a revised version of the manuscript that addresses the points raised during the review process.

We look forward to receiving your revised manuscript.

Kind regards,

Zhongliang Zu, Ph.D.

Academic Editor

PLOS ONE

Journal Requirements:

2. Please include captions for your Supporting Information files at the end of your manuscript, and update any in-text citations to match accordingly. Please see our Supporting Information guidelines for more information: http://journals.plos.org/plosone/s/supporting-information

Reviewers' comments:

Reviewer's Responses to Questions

**Comments to the Author**

1. Is the manuscript technically sound, and do the data support the conclusions?

Reviewer #1: Partly

Reviewer #2: Yes

2. Has the statistical analysis been performed appropriately and rigorously? 

Reviewer #1: Yes

Reviewer #2: Yes

3. Have the authors made all data underlying the findings in their manuscript fully available?

Reviewer #1: Yes

Reviewer #2: Yes

4. Is the manuscript presented in an intelligible fashion and written in standard English?

Reviewer #1: No

Reviewer #2: Yes

5. Review Comments to the Author

Reviewer #1: This study investigated the relationship between amide proton transfer-weighted (APTw) imaging signals with clinic-radiological findings in patients with cerebral infarction. Results on 29 patients showed the association of APTw imaging with clinical and prognostic features, including time after onset, lesion size, NIHSS, mRS and so on. The findings may facilitate better understanding of pathophysiology of ischemic stroke and thus benefit treatment strategies.

There are several issues that need to be addressed.

(1) As shown in Equation 1, the APTw effect is not only dependent on amide proton exchange rate of k, but also dependent on [amide proton]/[water proton] and T1. The studied 29 patients were at different stroke stages with varied time after onset (1.5-322 hours). I am not sure if [amide proton]/[water proton] and T1 still remain constant after hundred hours after stroke onset. If not, the APTw change may not be specific to pH alteration. It may be useful to provide images of T1w and T2w, as well as Z-spectra of infarct lesion and CNAWM for patients at different stroke stages.

(2) It is not clear why only 10th percentile of ADC was measured from DWI data. How about mean ADC? Is it correlated with APT?

(3) Please explain why a saturation power of 1.5 μT was employed. Was the CEST data corrected for B0 inhomogeneity? Please provide the details.

(4) Calculation of magnetization transfer ratio was described in the ‘Data processing’ section, but I did not see any results about it.

(5) How many slices for DWI? Why the acquisition matrix was 128x126 (rather than 128x128) and reconstructed to 256x256? Should DWI performed before APT imaging so that the slice with the largest ischemic lesions could be identified from DWI? Please re-order the imaging protocols.

(6) Please illustrate ROIs of CNAWM in Fig.2 and 3.

(7) In the discussion section, the authors stated that “we concerned mRS scores at the chronic phase were biased by pre-stroke mRS in this study”. What does this mean? Please clarify.

(8) Please keep consistent for MRA image illustration for Fig.2 and 3.

Reviewer #2: This manuscript retrospectively examined 29 stroke patients for association among their MRI markers and other clinical findings. In methodology, the histogram parameters including 10th, 25th, 50th, 75th, and 90th percentiles of APTW signal were calculated, and compared with other clinical factors including time after onset, lesion size, NIHSS and mRS. Furthermore, patients were subgrouped according to their prognosis and origins (cardioembolic v.s. no cardioembolic). Overall, the histogram analysis of APTw MRI is new for stroke patients. And the correlation with clinical factors were performed in detail and the results are sound. Therefore, I recommend for publication, after minor correction of following points.

1.The APT10, APT25 ... parameters are new to readers. Therefore, in writing, authors should emphasize and clarify the meaning of these parameters. For example, in method authors need to clarify APT10 is the “top” 10% since people can also measure the lowest 10%. In note of Table1 as well as in discussion，authors should clarify that APT 10 , APT 25 , ... are the 10th, 25th,... percentiles of APTW signal “within the ROIs”. Furthermore, authors may want to cite another paper on histogram analysis of CEST data (Liu T. et al “CEST MRI With Distribution-Based Analysis for Assessment of Early Stage Disease Activity in a Mouse Model of Multiple Sclerosis:An initial study”NMR Biomed 2019 Nov;32(11):e4139)

2.In figure caption(text) for Fig. 2 and Fig. 3, authors say both APTw signals in infarct area as indicated by DWI are lower compared with those for contralateral tissue. However, visually the lower signal lesions is not clear in Fig.3 APTw image. The authors should add more explanation on the comparison of Fig2 and Fig.3.

3.Line 42-43, in conclusion of the abstract, Authors claimed that "APTW imaging associates with clinico-radiological findings including the prognosis of patients with cerebral infarction."This is a bit over-interpreted, and the writing should be more accurate and specific.

4.APT25 (%), APT50 (%), APT75 (%) and APT90 (%) should be also reported both in Fig. 2 and Fig. 3.

5.MTRasym should be uniform throughout the paper. And the English writing should be more clear and accurate.

6. PLOS authors have the option to publish the peer review history of their article (what does this mean?). If published, this will include your full peer review and any attached files.

Reviewer #1: No

Reviewer #2: No

---

## [Author Response · Author response to Decision Letter 0]

7 Jul 2020

Reviewer #1: 

There are several issues that need to be addressed.

(1) As shown in Equation 1, the APTw effect is not only dependent on amide proton exchange rate of k, but also dependent on [amide proton]/[water proton] and T1. The studied 29 patients were at different stroke stages with varied time after onset (1.5-322 hours). I am not sure if [amide proton]/[water proton] and T1 still remain constant after hundred hours after stroke onset. If not, the APTw change may not be specific to pH alteration. It may be useful to provide images of T1w and T2w, as well as Z-spectra of infarct lesion and CNAWM for patients at different stroke stages.

RESPONSE: Thank you for this comment. As you pointed out, the APT effect (referred to as the APT ratio or APTR in the present study) depends on not only tissue pH, but also T1 relaxation time, water content, and amide proton content as shown in equations (1) and (2). 

However, it has been shown that APTR changes are largely determined by changes in pH (Zhou J, et al. Nature medicine. 2003;9). A previous study using phosphorus-31 MR spectroscopy has demonstrated that pH abnormalities of the infarct area remain, even in the subacute phase (Zollner JP, et al. Stroke. 2015;46).

T1 relaxation time and water content of the infarct area both increase over time during the acute to subacute phase, due to cytotoxic and vasogenic edema (Allen LM, et al. RadioGraphics. 2012;32). As the T1 relaxation time has a strong positive correlation with the water content in the infarct area (Lin et al. Journal of Cerebral Blood Flow and Metabolism 20;37), it seems likely that the two changes would cancel each other out, and thus contributions from both T1 and water content would be small. 

There has been no previous study focusing on dynamic changes of the amide proton content within the infarction. However, based on a study investigating amide proton content in peritumoral brain edema (Zhou J, et al. Magnetic resonance in medicine. 2003;50), it might decline in infarctions with cytotoxic edema, which might have affected the APTR to some extent.

As shown in equation (3), the APTW signal includes not only the APTR but also MTR’asym(3.5 ppm), which is attributed to the upfield NOE effect and does not reflect pH changes. However, a recent quantitative CEST study using animal models confirmed that the APTR accounts for a greater part of the MTRasym(3.5 ppm) changes than the NOE effect (Wu Y, et al. Magnetic resonance in medicine. 2018;79). Considering these facts, the APTW signal should primarily depend on tissue pH of the infarction, even in the subacute phase. We have added these points to the discussion (p.19–20, lines 370–388).

As requested, we have shown three cases with different stroke stages (hyperacute, acute, and subacute) in Fig 2–4 including T1WI, T2WI, Z-spectra, and MTRasym spectra of the infarction and CNAWM.

(2) It is not clear why only 10th percentile of ADC was measured from DWI data. How about mean ADC? Is it correlated with APT?

RESPONSE: Thank you for these questions. We used the ADC10 percentile because we initially thought it would better reflect the degree of cellular edema in infarctions.

We have calculated the correlation between the mean ADC values and APTW signals, and found that the mean ADC values were also positively correlated to the APTW signals (Table 4). Because the mean ADC value has been broadly used to measure the degree of diffusion restriction in infarctions, we have replaced the APT10 values with mean ADC values throughout the manuscript.

(3) Please explain why a saturation power of 1.5 μT was employed. Was the CEST data corrected for B0 inhomogeneity? Please provide the details.

RESPONSE: Thank you for these questions. The optimal saturation pulse strength for APTW imaging has not been established yet. Saturation pulse strengths of 0.5–3 μT have been used in previous studies. We selected a saturation power of 1.5 μT based on a previous animal study that reported the relationship between APTW signal and tissue lactic acidosis (Sun PZ, et al. Journal of cerebral blood flow and metabolism. 2007;27). However, a human study demonstrated that a saturation power of either 2 or 3 μT resulted in the better APTW image contrast between the infarction and the CNAWM than that of 1 μT (Zhao X, et al. Magnetic resonance in medicine. 2011;66). Considering these facts, we feel that the saturation pulse of 1.5 μT used in the present study was acceptable; however, slightly stronger saturation powers might have improved the APTW image contrast. We have added this point to the limitation section of the discussion (p.23, lines 450–459).

To correct the B0 susceptibility and inhomogeneity of the APTW MRI, we acquired a B0 map for off-resonance correction separately, with an identical spatial resolution to that of the APTW MRI. We have described this point in detail in the revised manuscript (p.7–8, lines 142–146).

(4) Calculation of magnetization transfer ratio was described in the ‘Data processing’ section, but I did not see any results about it.

RESPONSE: We apologize for the confusion. This study did not evaluate the MTR alone. Instead, we evaluated the MTR asymmetry (MTRasym) at ± 3.5 ppm as the APTW signal. We have rewritten the data processing section to clarify this (p.8, lines 147–164).

(5) How many slices for DWI? Why the acquisition matrix was 128x126 (rather than 128x128) and reconstructed to 256x256? Should DWI performed before APT imaging so that the slice with the largest ischemic lesions could be identified from DWI? Please re-order the imaging protocols.

RESPONSE: Thank you for these questions. The number of slices for DWI was 22.

With the 3.0-T MRI system (Achieva TX, Philips Medical Systems, Best, the Netherlands) we used, we were not able to directly set the acquisition matrix in k-space. Instead, the acquisition matrix was calculated backwards from the FOV, in-plane resolution, and sensitivity encoding factor; therefore, the number of phase-encoding steps did not result in just 128, but also 126. This is simply a matter of equipment operation and did not affect the ultimate images.

We generated a 256 x 256 matrix in DWI by filling the periphery of the 128 x 126 acquisition matrix in k-space with zeroes (zero filling interpolation: ZIP), which is a routinely-used interpolation technique (Bernstein MA, et al. Journal of Magnetic Resonance Imaging. 2001;14). We have described the number of slices and the zero-filling interpolation in the MRI data acquisition section of the materials and methods (p.7, lines 129–130).

In the present study, several MR sequences including transverse DWI, T2- and T1-weighted images, fluid attenuated inversion recovery, and 3D time-of-flight MR angiography were collected first. Subsequently, APTW MRI was conducted on the transverse slice with the largest ischemic lesions, which was selected with reference to DWI. We have clarified this point in the MRI data acquisition section and re-ordered the imaging protocols (p.6–8, lines 121–146).

(6) Please illustrate ROIs of CNAWM in Fig.2 and 3.

RESPONSE: Thank you for this suggestion. We have added the ROIs of the CNAWM to all the case presentations (Fig 2–4).

(7) In the discussion section, the authors stated that “we concerned mRS scores at the chronic phase were biased by pre-stroke mRS in this study”. What does this mean? Please clarify.

RESPONSE: We apologize for the confusion. A previous study demonstrated that a high pre-stroke mRS score was significantly associated with poor functional outcomes at the chronic phase (Quinn TJ, et al. Frontiers in neurology. 2017;8). Thus, we were concerned that the pre-stroke mRS score could be a confounding factor when evaluating the relationship between the APTW signal and the mRS score at the chronic phase. Therefore, we conducted an additional sensitivity analysis of a patient subpopulation comprising only those with low pre-stroke mRS scores (pre-stroke mRS ≤1), to exclude the influence of a high pre-stroke mRS score.

We have rewritten the corresponding part of the discussion to clarify this (p.21–22, lines 414–420).

(8) Please keep consistent for MRA image illustration for Fig.2 and 3.

RESPONSE: Thank you for this suggestion. We have presented all the MRA images as maximum intensity projections in Fig 2–4.

Reviewer #2 

1.The APT10, APT25 ... parameters are new to readers. Therefore, in writing, authors should emphasize and clarify the meaning of these parameters. For example, in method authors need to clarify APT10 is the “top” 10% since people can also measure the lowest 10%. In note of Table1 as well as in discussion，authors should clarify that APT 10 , APT 25 , ... are the 10th, 25th,... percentiles of APTW signal “within the ROIs”. Furthermore, authors may want to cite another paper on histogram analysis of CEST data (Liu T. et al “CEST MRI With Distribution-Based Analysis for Assessment of Early Stage Disease Activity in a Mouse Model of Multiple Sclerosis:An initial study”NMR Biomed 2019 Nov;32(11):e4139)

RESPONSE: Thank you for pointing this out. We obtained the 10th, 25th, 50th, 75th, and 90th percentiles of the APTW signals (%) within the ROI (APT10, APT25, APT50, APT75, APT90, respectively). The lowest 10% of the signal within an ROI was found below APT10, and the highest 10% was found above APT90. We have clarified this in the quantitative analysis of APTW images section (p. 9, lines 181–184). We have also emphasized that APTW signals were calculated within the infarction or CNAWM ROIs, throughout the paper.

We have cited the article by Liu et al. as an example of the application of APTW MRI in the introduction (p. 4, lines 71–73).

2.In figure caption(text) for Fig. 2 and Fig. 3, authors say both APTw signals in infarct area as indicated by DWI are lower compared with those for contralateral tissue. However, visually the lower signal lesions is not clear in Fig.3 APTw image. The authors should add more explanation on the comparison of Fig2 and Fig.3.

RESPONSE: Thank you for this suggestion. First, we have replaced Fig 3 with another case of subacute infarction, and added a case of hyperacute infarction in accordance with the suggestion by Reviewer #1. In addition, to make the figures easier to understand, we have added arrowheads indicating the infarct area to the APTW images (Fig 2–4).

3.Line 42-43, in conclusion of the abstract, Authors claimed that "APTW imaging associates with clinico-radiological findings including the prognosis of patients with cerebral infarction."This is a bit over-interpreted, and the writing should be more accurate and specific.

RESPONSE: Thank you for pointing this out. We have removed the description, and replaced it with the following conclusion: The poor prognosis group had a lower APTW signal than the good prognosis group. APTW signal was reduced in large infarctions, infarctions with low ADC, and in patients with high NIHSS and mRS scores. (p. 3, lines 43–46).

4.APT25 (%), APT50 (%), APT75 (%) and APT90 (%) should be also reported both in Fig. 2 and Fig. 3.

RESPONSE: Thank you for pointing this out. We have added APT25, APT50, APT75 and APT90 to the figure legends (Fig 2–4).

5.MTRasym should be uniform throughout the paper. And the English writing should be more clear and accurate.

RESPONSE: Thank you for pointing this out. We have standardized MTRasym throughout the manuscript. A professional English language editing company has checked and edited the manuscript. We have attached the separate file labeled 'Certificate of English Editing’ as proof that we have used the service.

---

## [Decision Letter · Decision Letter 1]

15 Jul 2020

PONE-D-20-12833R1

Correlations of amide proton transfer-weighted MRI of cerebral infarction with clinico-radiological findings

PLOS ONE

Dear Dr. Togao,

Thank you for submitting your manuscript to PLOS ONE. After careful consideration, we feel that it has merit but does not fully meet PLOS ONE’s publication criteria as it currently stands. Therefore, we invite you to submit a revised version of the manuscript that addresses the points raised during the review process.

We look forward to receiving your revised manuscript.

Kind regards,

Zhongliang Zu, Ph.D.

Academic Editor

PLOS ONE

Additional Editor Comments (if provided):

Please correct the reference 11.

Reviewers' comments:

Reviewer's Responses to Questions

**Comments to the Author**

1. If the authors have adequately addressed your comments raised in a previous round of review and you feel that this manuscript is now acceptable for publication, you may indicate that here to bypass the “Comments to the Author” section, enter your conflict of interest statement in the “Confidential to Editor” section, and submit your "Accept" recommendation.

Reviewer #1: All comments have been addressed

Reviewer #2: All comments have been addressed

2. Is the manuscript technically sound, and do the data support the conclusions?

Reviewer #1: Yes

Reviewer #2: Yes

3. Has the statistical analysis been performed appropriately and rigorously? 

Reviewer #1: Yes

Reviewer #2: Yes

4. Have the authors made all data underlying the findings in their manuscript fully available?

Reviewer #1: Yes

Reviewer #2: Yes

5. Is the manuscript presented in an intelligible fashion and written in standard English?

Reviewer #1: Yes

Reviewer #2: Yes

6. Review Comments to the Author

Reviewer #1: (No Response)

Reviewer #2: Reference 11 was cited at the wrong place. Currently it is cited in the introduction " or to detect subtle inflammatory changes at an early stage of multiple sclerosis [11]." This is not proper. Reference 11 not only examined APT signal, but also CEST signals at other offsets. Importantly, it applied a distribution-based analysis for CEST analysis. And in this paper the parameters APT10,APT15, ... in fact is also a quantitative desciption of APTw images. Therefore, the old reference 11 should be cited in the method within the "Quantitative analysis of APTW images" part

7. PLOS authors have the option to publish the peer review history of their article (what does this mean?). If published, this will include your full peer review and any attached files.

Reviewer #1: No

Reviewer #2: No

---

## [Author Response · Author response to Decision Letter 1]

17 Jul 2020

Reviewer #2:

Reference 11 was cited at the wrong place. Currently it is cited in the introduction " or to detect subtle inflammatory changes at an early stage of multiple sclerosis [11]." This is not proper. Reference 11 not only examined APT signal, but also CEST signals at other offsets. Importantly, it applied a distribution-based analysis for CEST analysis. And in this paper the parameters APT10,APT15, ... in fact is also a quantitative desciption of APTw images. Therefore, the old reference 11 should be cited in the method within the "Quantitative analysis of APTW images" part

RESPONSE: Thank you for pointing this out. We have removed the reference from the introduction and have cited it in the “Quantitative analysis of APTW images” part, as a previous instance applying a histogram-based analysis on CEST data (p. 9–10, lines 181–186).

---

## [Decision Letter · Decision Letter 2]

24 Jul 2020

Correlations of amide proton transfer-weighted MRI of cerebral infarction with clinico-radiological findings

PONE-D-20-12833R2

Dear Dr. Togao,

We’re pleased to inform you that your manuscript has been judged scientifically suitable for publication and will be formally accepted for publication once it meets all outstanding technical requirements.

Kind regards,

Zhongliang Zu, Ph.D.

Academic Editor

PLOS ONE

Additional Editor Comments (optional):

Reviewers' comments:

Reviewer's Responses to Questions

**Comments to the Author**

1. If the authors have adequately addressed your comments raised in a previous round of review and you feel that this manuscript is now acceptable for publication, you may indicate that here to bypass the “Comments to the Author” section, enter your conflict of interest statement in the “Confidential to Editor” section, and submit your "Accept" recommendation.

Reviewer #2: All comments have been addressed

2. Is the manuscript technically sound, and do the data support the conclusions?

Reviewer #2: Yes

3. Has the statistical analysis been performed appropriately and rigorously? 

Reviewer #2: Yes

4. Have the authors made all data underlying the findings in their manuscript fully available?

Reviewer #2: Yes

5. Is the manuscript presented in an intelligible fashion and written in standard English?

Reviewer #2: Yes

6. Review Comments to the Author

Reviewer #2: (No Response)

7. PLOS authors have the option to publish the peer review history of their article (what does this mean?). If published, this will include your full peer review and any attached files.

Reviewer #2: No